# Chronic bee paralysis as a serious emerging threat to honey bees

Giles E. Budge [1✉], Nicola K. Simcock[1], Philippa J. Holder[1], Mark D. F. Shirley[1], Mike A. Brown[2], Pauline S. M. Van Weymers [3], David J. Evans [3] & Steve P. Rushton[1]

Chronic bee paralysis is a well-defined viral disease of honey bees with a global distribution that until recently caused rare but severe symptomatology including colony loss. Anecdotal evidence indicates a recent increase in virus incidence in several countries, but no mention of concomitant disease. We use government honey bee health inspection records from England and Wales to test whether chronic bee paralysis is an emerging infectious disease and investigate the spatiotemporal patterns of disease. The number of chronic bee paralysis cases increased exponentially between 2007 and 2017, demonstrating chronic bee paralysis as an emergent disease. Disease is highly clustered spatially within most years, suggesting local spread, but not between years, suggesting disease burnt out with periodic reintroduction. Apiary and county level risk factors are confirmed to include scale of beekeeping operation and the history of honey bee imports. Our findings offer epidemiological insight into this damaging emerging disease.

[1] School of Natural and Environmental Sciences, Newcastle University, Newcastle upon Tyne, Tyne and Wear NE1 7RU, UK. [2] National Bee Unit, Animal and Plant Health Agency, Sand Hutton, York YO41 1LZ, UK. [3] Biomedical Sciences Research Complex and School of Biology, University of St Andrews, North Haugh, St Andrews, Fife KY16 9ST, UK. ✉email: giles.budge@newcastle.ac.uk

Animal pollinators are necessary for the reproductive success of 88% of flowering plants globally[1] and contribute to the yield and quality of many crops[2,3]. Managed Western honey bees (*Apis mellifera*) offer mobile pollination services to complement wild pollinators and account for 30–50% of this ecosystem service[4,5]. Regional declines have been reported in both honey bee[6] and wild pollinator populations in the face of multiple interacting pressures that include land-use intensification, agrochemical exposure and the impact of parasites/pathogens[7,8].

Emerging infectious diseases (EIDs), defined as newly appearing in a population or rapidly increasing in incidence or geographic range[9], often arise from livestock or plant movements[10]. Owing to their use for managed pollination and honey production, the global trade in honey bees has expanded massively[11]. This trade has the known potential to also increase the geographic distribution of viral, bacterial and fungal honey bee parasites and pathogens. Consequently, this expansion has also been associated with an increased prevalence of EIDs, some of which have been implicated in large-scale population (colony) losses. Host jumps from the Eastern honey bee (*Apis cerana*) by the ectoparasitic mite *Varroa destructor*[12,13] and the microsporidium *Nosema ceranae*[14–16] have resulted in the emergence of new honey bee diseases of the Western honey bee and the loss of millions of colonies worldwide. There is growing evidence that many pathogens found in honey bees are shared between other pollinator species[17,18], providing further opportunity for pathogens to spread to new biogeographical areas.

Chronic bee paralysis virus (CBPV) is an unclassified bipartite RNA virus[19] that until recently caused a rare but severe chronic paralysis disease in honey bees, with very characteristic symptoms including abnormal trembling, flightlessness and shiny, hairless abdomens[20]. Infected symptomatic individuals die within a week[21,22] leading to mounds of dead bees outside affected colonies, which sometimes collapse[23] or are too weakened for pollination or honey production. Chronic bee paralysis has a worldwide distribution, with recent increased incidence of CBPV reported in Asia, Europe and North America[24–26]. Historic UK disease prevalence was reported as 2% in 1966, but data were generated from records of samples submitted to a pathology laboratory for diagnosis, and so are likely skewed[27].

Honey bee health has been monitored in the UK by a government-funded apiary inspection programme run by the National Bee Unit (NBU)[28]. Since 2006, data from apiary visits have been collated into a national database known as BeeBase, which includes metadata about locality, colony health and free text fields to record any anomalies or non-statutory disease. In this study, we obtain observations of chronic bee paralysis from BeeBase to investigate the spread in English and Welsh apiaries over 12 years, using a combination of epidemiological analyses of the patterns of disease based on the spatiotemporal distribution of outbreaks in apiaries and in counties. We use this analytical framework to test the hypothesis that chronic bee paralysis is an emergent disease, quantify the spatial dependency of risk and investigate county-level risk factors associated with the underlying epidemiology.

## Results

**Disease prevalence**. A total of 79,873 apiary visits to 24,186 beekeepers were conducted by government bee health inspectors during the period 2006 to 2017 (Supplementary Fig. 1). The majority of visits occurred during the active beekeeping season, between April and September (Supplementary Fig. 2). Although most visits were initiated by NBU inspectors (non-call-outs ~82% of visits), some were conducted in response to call-outs by beekeepers with concerns about honey bee health (~18% of visits;

Supplementary Fig. 2). Interestingly, the frequency of call-out visits to amateur beekeepers (<40 colonies) was about double those of professional beekeepers with ≥40 honey bee colonies, suggesting professional beekeepers were less likely to call out a NBU inspector.

There were no cases of chronic bee paralysis in 2006. However, the proportion of apiary visits where chronic bee paralysis was recorded rose exponentially between 2007 and 2017 ($n = 11$, $t = 10.07$, $P = 1.50e{-}06$; Fig. 1a). The rate of increase differed between professional and amateur beekeepers, with 1.98 times more disease found in apiaries owned by professional beekeepers ($n = 11$, $t = 2.668$, $P = 0.0144$), although much of this difference occurred from 2014 onwards (Fig. 1b). Interestingly, there was an increase in the number of chronic bee paralysis cases regardless of whether or not apiary visits were at the request of the beekeeper. In 2007, chronic bee paralysis was only recorded in one English county; however, by 2017, it was recorded in 39 of the 47 English and 6 of the 8 Welsh counties.

**Confirmation of CBPV**. In 2017, NBU inspectors and bee farmers were asked to collect symptomatic adult honey bees from colonies they believed to be showing symptoms of chronic bee paralysis in the field, as well as returning foragers from 24 asymptomatic colonies. Bees were tested for the presence of CBPV using an established reverse transcription quantitative polymerase chain reaction (RT qPCR) assay[29]. Overall, CBPV was detected in symptomatic honey bees from 21/24 colonies. Interestingly, CBPV was detected in 7/23 colonies reported to be asymptomatic. No template controls and blank extractions tested negative for CBPV, and the calibration curves for CBPV ($R^2 = 0.997$; Intercept = 28.99; Slope = −3.86) and honey bee 18S ($R^2 = 0.998$; Intercept = 21.38; Slope = −3.31) facilitated relative quantification of CBPV-positive samples. When quantitative results were normalised to the mean of CBPV in asymptomatic colonies, the quantity of virus was over 235,000× higher in the symptomatic group (Fig. 2) compared to the asymptomatic group (Source Data). These data confirm CBPV in the majority of survey cases and also suggest a false positive rate of approximately 12% in our survey data.

**Disease clustering**. If chronic bee paralysis was spreading contagiously, then we might expect cases in apiaries to be clustered spatially and through time. Values of the $K$ function, which indicates case to case proximity, were higher than that expected by chance at all inter-apiary distances when the data were combined for all years (Fig. 3; Source Data), indicating that the pattern of disease was significantly clustered at all distances up to 40 km. Separating the data into chronic bee paralysis cases in individual years indicated that spatial clustering was also present in each individual year.

However, there was no significant space–time clustering. This suggests that, while there was spatial clustering within years, this was not continued through time to subsequent years. Annual kernel smoothing plots of disease density for England and Wales shows that cases of chronic bee paralysis occurred in pockets distributed across England and Wales but that the position of clusters changed from year to year (Fig. 4). This supports the formal analyses of space–time clustering by suggesting that cases of disease appeared and spread spatially each year but that clusters did not persist between years.

**Apiary-level disease risk factors**. Identifying risk factors for chronic bee paralysis at the level of the apiary was difficult because there was not an appropriate error model with which to analyse the data. The disease was comparatively rare (zero

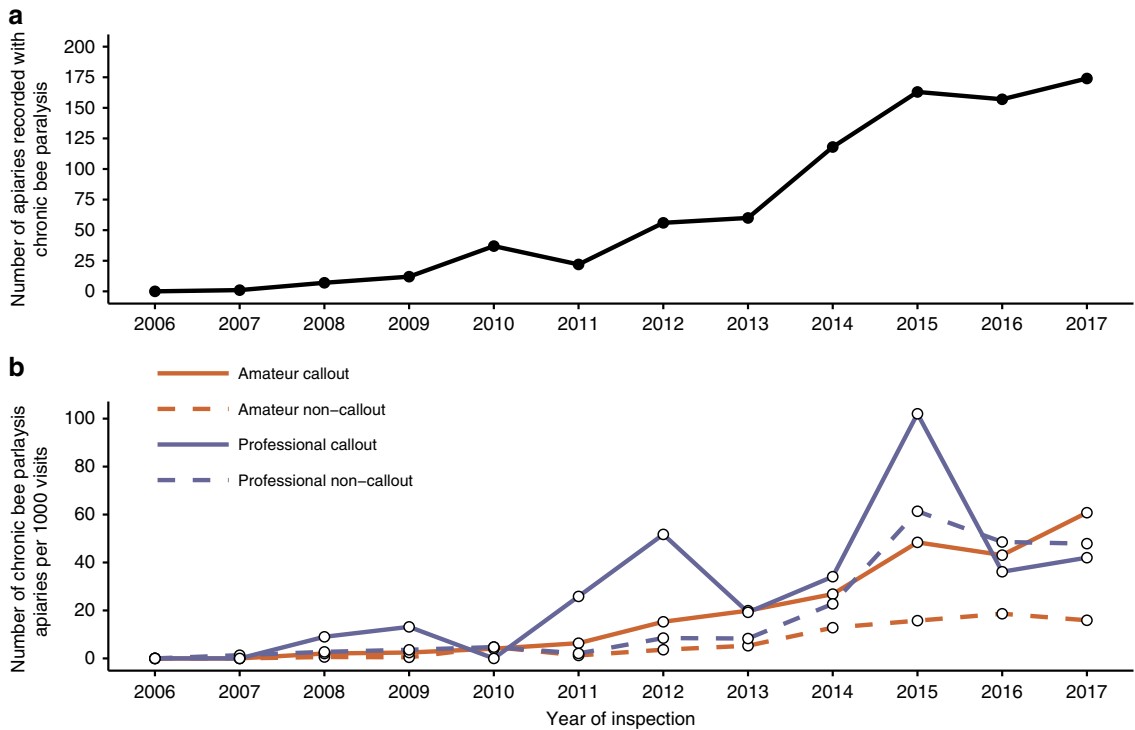

**Fig. 1 Chronic bee paralysis cases in England and Wales. a** Number of visited apiaries recorded with chronic bee paralysis between 2006 and 2017. **b** The number of apiaries (per 1000 visits) where chronic bee paralysis was recorded for amateur and professional beekeepers. Data are separated into visits that occurred because of a call-out by the beekeeper or not (non-call-out).

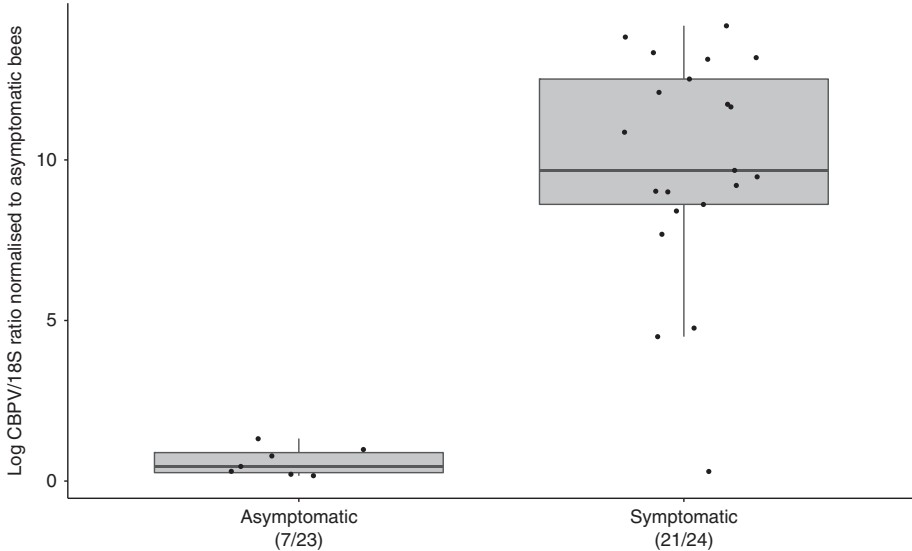

**Fig. 2 Box and whisker plot showing real-time RT qPCR testing for the presence of chronic bee paralysis virus in 24 symptomatic and 23 asymptomatic colonies.** All data points are presented. Virus quantity is expressed relative to the mean amount of virus detected in asymptomatic samples, demonstrating far higher quantities of chronic bee paralysis virus in adult bees from symptomatic colonies.

inflated) and there was obvious spatial clustering. Linear models with normal, Poisson or binomial errors led to residuals that were over-dispersed. When analysed using generalised estimating equations (GEE), which are more robust to dependence between observations, risk of chronic bee paralysis in apiaries increased significantly with year ($n = 64{,}806$; $z = 19.64$). This increase was dependent on whether the beekeeper was a professional or an amateur ($n = 64{,}806$; $z = 5.4$), and whether or not the beekeeper had imported honey bees in the 2 years prior to apiary visit ($n = 64{,}806$; $z = 5.7$). Apiaries owned by professionals had a 1.5 (confidence intervals (CI) 1.4–1.6) times greater risk of recording chronic bee paralysis than those owned by amateurs. In addition, beekeepers who imported honey bees 2 years prior to apiary visit had a 1.81 (CI 1.68–1.96) times greater risk of chronic bee paralysis than those who did not. Risk of chronic bee paralysis being recorded at apiary visit increased by a factor of 1.25 per year (CI 1.24–1.26) across England and Wales between 2007 and 2017, reflecting the emergent status of the disease.

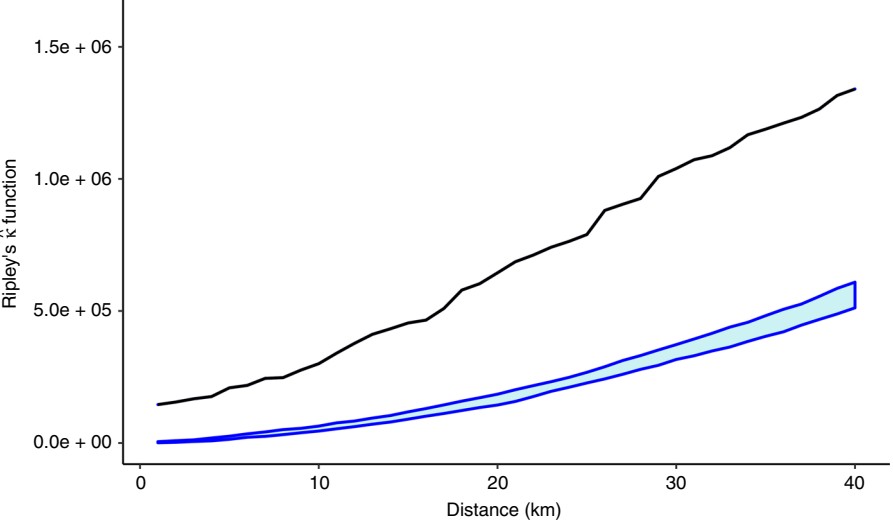

**Fig. 3 K function analysis showing the extent to which cases of chronic bee paralysis are clustered spatially for all years (2007–2017).** The observed $\hat{k}$ (black line) represents a mean count of the number of chronic bee paralysis cases within fixed distances of cases (x axis). Higher values of $\hat{k}$ show stronger clustering. Upper and lower 95% confidence intervals are shown for estimates of $\hat{k}$ derived by allocating cases to randomly selected apiary sites and repeating 20 times (blue lines). Since the observed $\widehat{k}$ values (black line) are substantially greater than those derived from random resampling at all distances, we can conclude that cases of chronic bee paralysis are nearer to each other than we would expect by chance and therefore clustered. Note that the calculation of $\hat{k}$ also includes an adjustment for coastal edge effects.

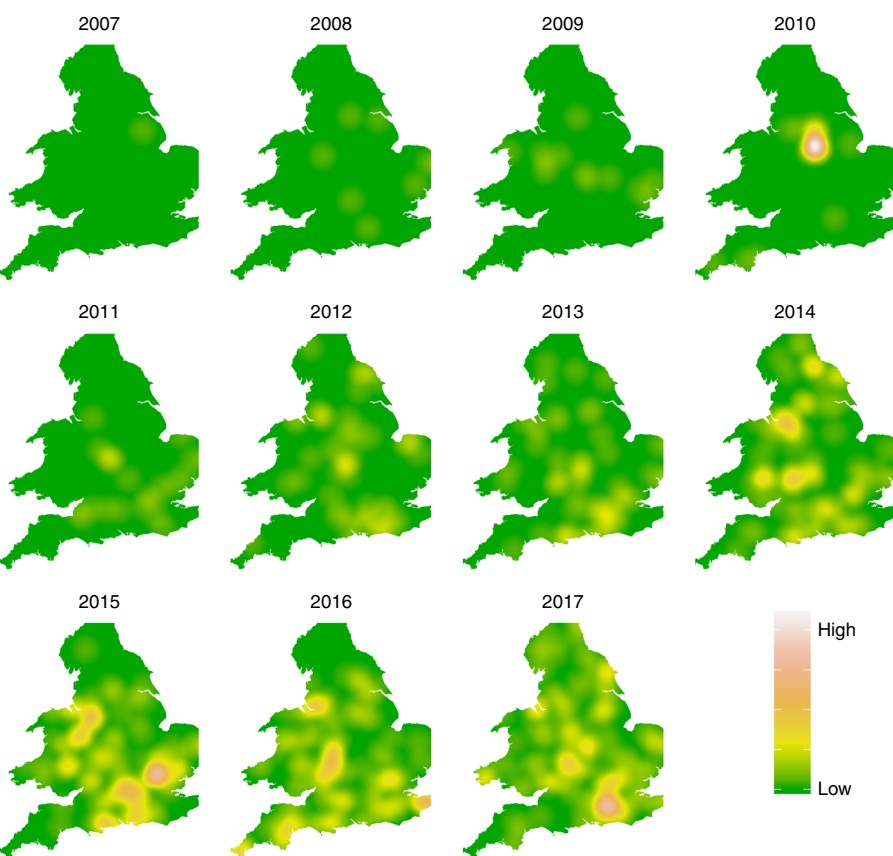

**Fig. 4 Kernel density maps showing the intensity of chronic bee paralysis in England and Wales between 2007 and 2017.** The pattern of cases indicates that disease clusters do not appear in the same positions year on year. The county boundaries for England and Wales were sourced from the Database of Global Administrative Areas (http://GADM.org).

**Area-level honey bee import data.** In total, there were 130,746 honey bee imports between 2007 and 2017, mainly comprising queen imports (90%) with a tendency for an increasing number of imports over time (Supplementary Table 1). Honey bee imports originated from 25 different countries: Greece $n = 43,057$; Slovenia $n = 21,853$; Italy $n = 14,591$; Hawaii $n = 10,231$; Denmark $n = 7391$; New Zealand $n = 7279$; Cyprus $n = 5809$; Romania $n = 4658$; Czech Republic $n = 2643$; Germany $n = 2590$;

Argentina $n = 1970$; France $n = 1816$; Spain $n = 1696$; Poland $n = 1682$; Malta $n = 1141$; Hungary $n = 422$; Austria $n = 303$. An additional 1614 honey bee imports arrived from Australia, Ireland, Croatia, Netherlands, Portugal, Sweden, Luxemburg and Estonia (Table 1).

For the seven countries from which >5000 honey bee imports were recorded, the proportion of direct imports that could be allocated to county varied, with imports from Denmark mainly direct to beekeepers (64%), whereas those from Greece were mainly delivered to commercial operations (77%; Fig. 5).

**Area-level risk factors**. The relative risk of chronic bee paralysis rose over time and varied between counties (Fig. 6; Source Data). There was a significant relationship between the log-transformed number of cases per county and time (Generalised Linear Model; $n = 605$, $t = 14.353$, $P = <0.001$). The log-transformed number of cases per county was also significantly related to the number of imported honey bees from Denmark (Generalised Linear Model; $n = 605$, $t = 3.826$, $P = <0.001$); however, the effect was small with a unit increase in honey bee imports leading to an increase in 3.5 cases per thousand apiaries (CI 1.53–5.50). No significant relationship was shown between chronic bee paralysis cases and the number of imported honey bees from Slovenia, Italy, Greece, Hawaii, New Zealand or Cyprus.

The Deviance Information Criterion (DIC) for the intercept of a Besag–York–Mollie model (BYM) using observed and expected cases of chronic bee paralysis was 2688 when assuming no dependency on time or importation of honey bees. The addition of visit year as a covariate to the model reduced the DIC to 2520. Addition of the number of apiaries receiving honey bee imports from Denmark in each year reduced the DIC further to 2516. Analysis of the posterior means for the final BYM model indicated that there was a 32.9% increase in cases of chronic bee paralysis per year (credible intervals 25.2–40.6), and this rate was increased by a further 3.5% (credible intervals 1.51–5.39) when honey bee imports from Denmark were included.

**Discussion**

Our data clearly indicate that since 2007 there has been an exponential increase in the number of cases of chronic bee paralysis in honey bee apiaries across England and Wales, providing the first report of recent disease emergence. Our RT qPCR data confirmed the presence of CBPV at high levels in symptomatic adult honey bees from the majority of disease cases and at low levels in some asymptotic bees, which is consistent with previous reports of chronic bee paralysis[29,30]. RT qPCR data suggested that the false positive rates from using observational survey data in the absence of virus confirmation were low. Virus replication can be triggered by parasite coinfection or agro-chemical exposure, which may hinder bee antiviral mechanisms[31]. Interestingly, there have been numerous recent reports of increased incidence of the virus that causes chronic bee paralysis from around the world. In the US, CBPV was detected at 0.7% prevalence in 2010 but has more than doubled annually to reach 16% in 2014[24]. CBPV incidence increased in Italy from 5% in 2009 to 10% in 2010[26], and China has seen more remarkable increases from 9% to 38%[25]. Taken together, these observations suggest an increase in the incidence of the causal agent of chronic bee paralysis in Asia, Europe and North America, supporting our field observations of the emergence of chronic bee paralysis across England and Wales.

We are the first to report that chronic bee paralysis showed clear clustered patterns within year with significant clustering up to 40 km (Figs. 2, 3). CBPV is capable of multiple routes of

**Table 1 Summary statistics for generalised linear models investigating trends in the number of counties (response variable) receiving honey bee imports through time (predictor variable) from different countries of origin.**

| Country | Estimate | Std. error | t value | P value |
|---|---|---|---|---|
| Cyprus | 0.1000 | 0.4816 | 0.208 | 0.84013 |
| Denmark | 2.5818 | 0.3876 | 6.660 | 9.26e−05 |
| Greece | 0.2364 | 0.3970 | 0.595 | 0.56627 |
| Hawaii[a] | −1.7818 | 0.3766 | −4.731 | 0.001072 |
| Italy | 1.9273 | 0.3021 | 6.380 | 0.000128 |
| New Zealand | 0.3727 | 0.2235 | −1.668 | 0.12970 |
| Slovenia | 2.6460 | 0.446 | 5.932 | 0.00022 |

These analyses highlight a significant increase in the number of counties receiving imports from Slovenia, Italy and Denmark between 2007 and 2017 ($n = 11$ for each GLM).
[a]Imports from Hawaii ceased in 2010 after the discovery of the small hive beetle (*Aethina tumida*).

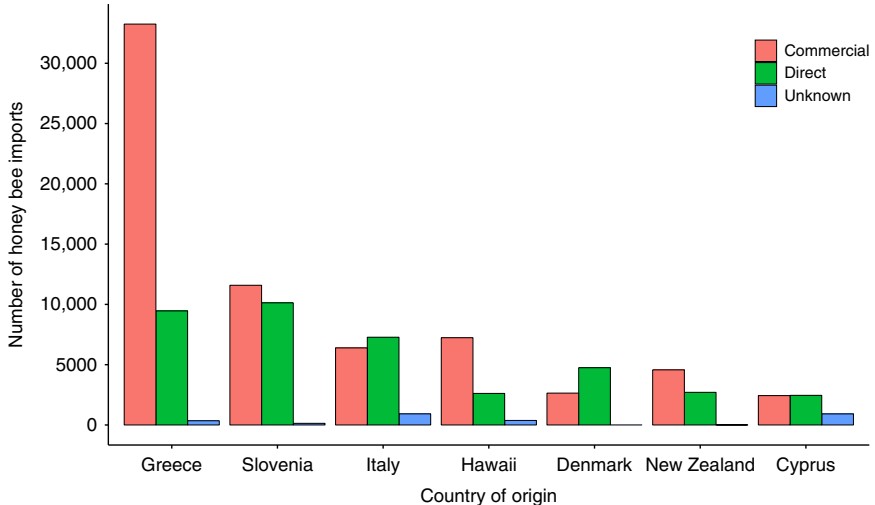

**Fig. 5 Total number of honey bee imports between 2007 and 2017.** Imports are allocated directly to beekeepers (green), allocated to commercial importers (pink) or of unknown allocation due to missing data (blue).

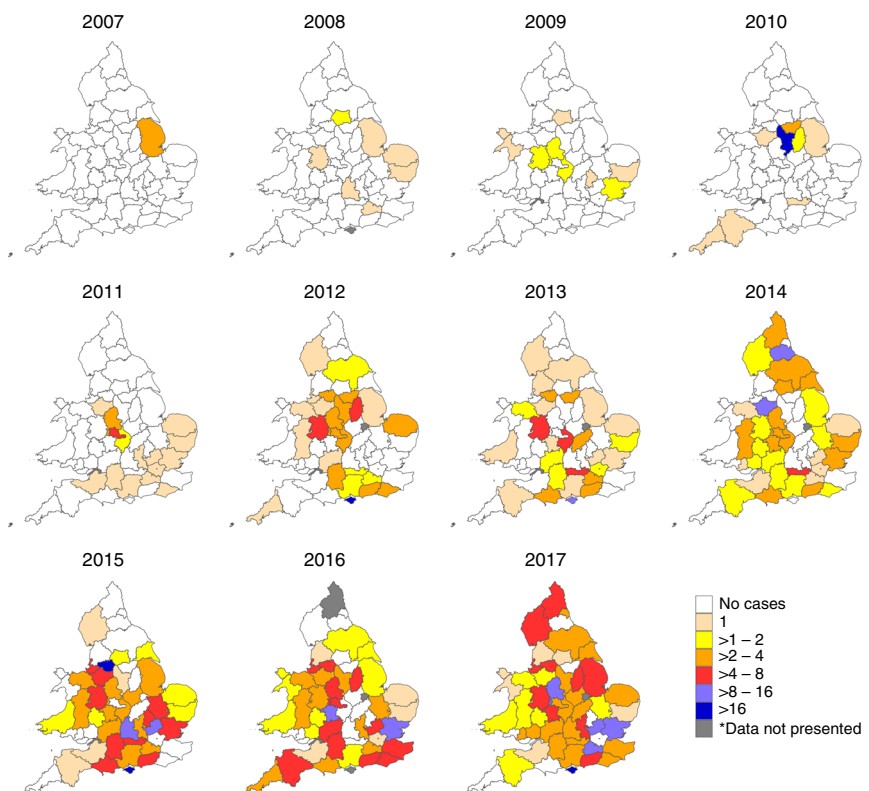

**Fig. 6 Relative risk of chronic bee paralysis for each county between 2007 and 2017 compared to the overall expected risk.** The overall expected risk (all cases in all years divided by the total visits in all years) was multiplied by the number of visits per county. The relative risk represents a fold change in disease risk compared to the expected. *To maintain confidentiality, data are not presented when fewer than five apiaries were visited in a single county/year. The county boundaries for England and Wales were sourced from the Database of Global Administrative Areas (http://GADM.org).

transmission within the hive, including faecal oral[32], mechanical[27] and possibly vertical transmission through the queen[23]. Infected adult bees carry the replicating virus for up to 6 days before showing symptoms[21], raising the possibility of infective individuals visiting shared forage sites or moving the virus by robbing distant honey bee colonies. However, such transmission routes can only operate at a spatial scale of twice the maximum foraging distance of a honey bee, typically <10 km in the UK[33], and could therefore not account for the larger-scale clustering we observed. Beekeeping behaviours have been implicated in movements of other honey bee diseases[34], and one quarter of apiarists had multiple apiary sites, and therefore operate at a higher spatial scale than a single apiary location. Our data therefore suggest a combination of local spread between apiaries within years as well as spread at a scale of beekeeping operation. Interestingly, we found no spatiotemporal clustering, indicating that areas with a high disease burden were generally not the same between years, perhaps indicating that chronic bee paralysis may burn out in a beekeeping season, before being reintroduced in new areas in subsequent years.

The area-based models not only allowed for investigation of the impacts of management and honey bee imports at a large scale but also allowed for the inclusion of spatial dependence associated with diseased areas adjoining others where the disease was found. Here it is evident from the wide spread of early cases in 2008 that the pattern of disease does not follow a typical invasion front, associated with introgression of disease at a focal point followed by diffusive spread away from the focus. A natural corollary to this is that, if introgression of disease and subsequent spread is the mechanism by which chronic bee paralysis has gained emergent disease status, then it must have occurred multiple times at widely distributed points across England and

Wales. This does not suggest a natural invasion process. Our results indicate that the risk of chronic bee paralysis being recorded during apiary visits was 1.5 (CI 1.4–1.6) times greater for professional beekeepers when compared to amateurs. Previous cases of chronic bee paralysis in England have been positively correlated with colony density[27], which could explain the link with professional beekeepers who naturally have a higher average number of colonies on an apiary site (12.9) compared to amateurs (4.9). Many management practices will differ between amateur and professional groups, and some, such as the addition of pollen traps, are known to induce chronic bee paralysis symptoms[35].

Our results also highlight that the risk of chronic bee paralysis being recorded during apiary visits was 1.81 (CI 1.68–1.96) times greater for beekeepers who had imported honey bees 2 years prior to visit compared to those who did not import honey bees. Imports from Denmark also accounted for a modest 3.5 cases per 1000 imports. UK beekeepers frequently import honey bee stocks from abroad and have done so for hundreds of years. The patterns of importation over the period studied varied between years (Supplementary Table 1), and our results indicate that the level of importation of honey bees from abroad was a contributory feature to the spatial and temporal pattern of disease. The importation of bees to apiaries does not happen on a yearly basis in the UK, where honey bee queens are replaced as they become less productive after about 2 years. This periodic introduction of imported honey bees may partially explain why the disease gets a short-lived foothold in different areas before burning out and appearing elsewhere in different counties. This leads to two possible hypotheses. First, imported honey bees after 2006 were carriers of chronic bee paralysis or a new more virulent strain thereof. Second, honey bees imported after 2006 were susceptible to the resident strains of CBPV to which they were not exposed in

their source country. It is not possible to distinguish between these two hypotheses with the current state of research.

Our attempts to attribute apiary-level risks was made difficult due to there being no appropriate error models with which to analyse disease data. The data were zero-inflated and residuals for all linear models attempted were over-dispersed. Zero-inflated data tend to have too few non-zero responses, a common occurrence when disease is rare, and occur when there is more than one set of epidemiological processes involved. Typically, one set of conditions is responsible for whether or not disease can occur and another set determining the local magnitude or spread[36]. In the case of chronic bee paralysis, emergence probably arose from a combination of an initial source or risk in the landscape (such as imported honey bees) followed by processes that allowed local apiaries to be colonised, such as local contact through management or weather. Zero-inflated regression models aimed at identifying risk factors for the two sets of processes did not converge. Furthermore, models assuming other distributions for the residuals were over-dispersed, indicating that error models were inappropriate or the models had missing predictor variables[37]. In the present context, we could hypothesise that the local clustering of disease would lead not only to zero-inflation but also to aggregation across local apiaries, processes not easily linked in a linear modelling framework.

We present the first epidemiological study of chronic bee paralysis and demonstrate that the disease is emergent in England and Wales. Our data present correlatory evidence linking disease emergence with the international movement of honey bees, highlighting a transnational risk. Future experimental work should challenge our postulated hypotheses to determine the drivers of emergence. Clearly something has changed, and experiments assessing the virulence of different CBPV genotypes, the susceptibility of different honey bee races and the contrasting management practices of professional and amateur beekeepers will help discover the drivers of the current disease emergence. These studies should also identify beekeeping management practices that help reduce or mitigate the damage caused by this severe EID of honey bees.

## Methods

**Disease data.** The Department for Environment, Food and Rural Affairs (Defra) and the Welsh Government have long established honey bee health monitoring systems operated by the NBU (see www.nationalbeeunit.com). Trained honey bee health inspectors operate regional disease control and beekeeper training programmes primarily for the management of statutory foulbrood diseases[28]. Bee health inspectors operate a prioritised risk-based inspection service using a combination of the prior history of disease and the risk of exotic pest incursion[28]. Bee health inspectors gather data on colony health at each apiary visit, including colony size, management issues such as queen problems, whether the beekeeper is registered as a honey bee importer and the presence of both statutory and non-statutory diseases. All observations are temporally and spatially explicit, and since 2006, all apiary visit data have been collated into a national database known as BeeBase. We assigned chronic bee paralysis cases in BeeBase by including those colonies specifically marked as having chronic bee paralysis using a check box and by searching the colony notes field for certain keywords. Keywords included those indicating the presence of the disease (paralysis, cpv, cpbv, cbp or bpv), symptoms of paralysis (shivering, shaking, quivering and trembling) or other associated symptoms (black, shiny, crawling, many dead bees, k-wing). Each putative case was individually checked to ascertain the likely presence of chronic bee paralysis. Included cases had to mention the disease by name or mention paralysis symptoms as well as at least one additional associated symptom.

**Testing for CBPV.** Our visit data represented historic cases of chronic bee paralysis, and so it was not possible to confirm the presence of CBPV in all cases. Instead, we surveyed adult bees from 24 colonies deemed symptomatic by NBU inspectors and bee farmers in 2017 and compared virus levels to adult bees from 23 colonies deemed asymptomatic. Adult bees showing symptoms of paralysis were sampled from symptomatic colonies and healthy returning foragers were sampled from asymptomatic colonies. Each sampled bee was immediately immersed in RNAlater, posted for overnight delivery and frozen at −80 °C upon receipt.

For each sample type (asymptomatic or symptomatic), an individual honey bee was ground to homogeneity using a Precellys tissue homogeniser (Bertin Instruments) with ~0.5 mL of 2.3 mm zirconia silica beads (BioSpec #11079125z) and 300 μL lysis buffer (20 μL β-mercaptoethanol in 280 μL lysis buffer, GeneJet Kit, thermofisher). RNA was extracted using a Genejet Kit (Cat. no.: K0731) following manufacturer's instructions.

Testing for CBPV was achieved using CBPV_F (CGCAAGTACGCCTTGAT AAAGAAC), CBPV_R (ACTACTAGAAACTCGTCGCTTCG) and dual labelled probe CBPV_T (TCAAGAACGAGACCACCGCCAAGTTC) designed to the RNA-dependent RNA polymerase gene[29]. The RNA quality within each sample was normalised using AJ307465-955F (TGTTTTCCCTGGCCGAAAG), 1016R (CCCCAATCCCTAGCACGAA) and dual labelled probe 975T (CCCGGGTAACC CGCTGAACCTC) designed to *A. mellifera* 18S rRNA, and qPCR was performed using the relative standard curve method[38]. Both fluorogenic probes were modified 3′ with TAMRA (tetra-methylcarboxyrhodamine) and 5′ with FAM (6-carboxyfluorescein). Real-time reactions were set up in duplicate using hydrolysis probe (TaqMan®) chemistry in 96-well reaction plates using iTaq Universal Probes 1-Step Kit (Bio-rad, Cat no: 1725141), according to the manufacturer's protocols. Each reaction comprised 12.5 μL iTaq universal probes reaction mix (2×), 0.625 μL iScript advanced reverse transcriptase, 1 μL (7.5 μM) of each Forward and Reverse primer, 0.5 μL (5 μM) probe and 1 μL of RNA in a final volume of 25 μL. Reactions were carried out within the LightCycler 96 (Roche Life Sciences) and following the published cycling protocols[29,38]. LC96 software was used to produce qualitative positive/negative calls and determine the quantification cycle ($C_q$) for each sample following the User Training Guide, Version 2.0 (Roche Life Sciences). Both duplicate wells were required to have CBPV-positive calls for a sample to be deemed positive.

**Disease clustering.** The analysis of spatial and temporal clustering seeks to challenge the hypothesis that outbreaks of disease in apiaries are clustered together in time and space more than expected by chance. $K$ function analysis is a technique that analyses disease clustering by estimating the proximity of cases to each other and comparing observed counts of outbreaks to those that would occur by chance[39]. $K$ functions for space–time do the same but consider time as an additional dimension separating cases, providing insight into the persistence and temporal spread of clusters. We used a modification of the $\hat{k}$ function routine in the *splancs* package of R[40,41] to estimate the extent to which chronic bee paralysis cases were clustered in space and time. Typically, $K$ function analysis compares the proximity of cases to those that might occur with complete spatial randomness. However, apiaries are not distributed randomly so we used the known distribution of all visited apiaries as our population from which random samples could be drawn, to represent possible locations where there was no disease. The $K$ function calculates a measure of the expected number of disease events (cases) within a given distance of an arbitrary event after adjusting for edge effects that might occur, for instance, at the edge of a study area (the coastline of England and Wales), beyond which apiaries would not be found. We compared the estimated $K$ function of chronic bee paralysis events over a range of distances (1–40 km) with that derived from random draws of an equivalent sample size from the known locations of all visited apiaries. We assessed the significance of clustering on the extent to which the observed $K$ values over all distances lay outside the envelope derived from the simulated maximum and minimum $K$ values derived from sampling the control apiaries. Elevated values for $K$ outside the ranges of values calculated from repeated randomly selected sets of apiaries indicate spatial clustering. We undertook spatial analyses for each individual year before extending to space–time $K$ function, with a time step of 1 year to assess clustering in time and space. We used kernel density smoothing within the *splancs* package to produce annual smoothed representations of the incidence of disease across England and Wales to illustrate the pattern of disease through time.

**Apiary-level disease risk factors.** We analysed the potential role of husbandry-associated risk factors for disease within each apiary. Since the data were over-dispersed, zero-inflated and there was potential for clustering of disease, we investigated the impacts of apiary-level risk factors on disease using GEE, which allow for serial dependence in binomial outcomes[42] using the *gee* package in R[43]. We analysed the disease risk associated with small-scale (amateur) apiarists owning <40 colonies or professionals owning ≥40 colonies and also whether beekeepers had imported honey bees (see below) in the 2 years prior to the apiary visit.

**Area-level honey bee import data.** The apiculture sector in England and Wales comprises predominantly (>30,000) amateur beekeepers, with relatively few professional beekeepers (~400) who keep >40 honey bee colonies[28]. Both the amateur and professional beekeeping communities have some dependency on imported honey bees from EU and non-EU countries. Imports frequently arrive as queens with a few attending workers. Upon arrival, queens are introduced into honey bee colonies where it takes several months for the offspring of the imported queen to replace those from the old queen. Imports can also arrive as package bees that contain a queen with a few thousand workers but no brood or nucleus colonies that represent small colonies containing all life stages. We obtained country-specific records of honey bee imports from 2007 from BeeBase and summed all import

types. We sought to distinguish between beekeepers who import honey bees for their own use and those who import honey bees commercially for subsequent distribution to other beekeepers. We allocated imports direct to beekeeper apiary location(s) when the sum of imports in any 1 year was less than the maximum number of colonies kept using BeeBase records (referred to as direct imports). Where the number of imports were greater than the maximum number of colonies kept, we classified these imports as commercial imports. To account for the delay between importation and the establishment of a honey bee colony headed by an imported queen, we created a 2-year lag function whereby honey bee imports in 2007 were not accounted for until 2008 but then carried into 2009 to represent the average productive lifespan of a honey bee queen.

**Area-level disease risk factors.** Spatial dependency in the incidence of disease can be analysed at a coarser scale by considering how cases are distributed in areas (counties) rather than as points (apiaries). The county boundaries for England and Wales were sourced from the Database of Global Administrative Areas (http://GADM.org). To visualise disease risk over time, we calculated the overall expected county-level risk between 2007 and 2017 as the total number of chronic bee paralysis-positive visits divided by the total number of visits, multiplied by the annual number of visit per county and compared this to the actual number of positive visits expressed as a relative risk.

We then undertook two modelling approaches to investigate how chronic bee paralysis spread in counties through time in relation to the history of country-specific honey bee imports.

First, we investigated the rate of disease spread in counties using linear mixed-effect models with the number of apiaries with chronic bee paralysis in each year/county as a response variable; year, the total number of apiaries present in the county and the lagged number of country-specific honey bee imports as independent variables and county as a random effect. Second, we used the BYM model, which includes the spatial dependency between adjacent areas such that the risk of disease in any one area is also related to the pattern of disease in those areas that are adjacent to it[44]. The total number of chronic bee paralysis cases and the total number of visits in England and Wales were used to calculate the expected number of cases in a county given the total number of visits in it. We used observed cases of chronic bee paralysis in a county as the dependent variable, the number of expected cases as an offset and contiguity between counties as a measure of the likely spatial dependence in the response. We extended this model to include a covariate for importations of bees from countries in each year. Given that importations over the study period were sparse for some countries, we used a cut off and only modelled those countries for which importations in excess of 5000 honey bee imports were made across the study period. We compared models with covariates against null models with no explanatory variables using the DIC by using the *inla* package in R[45].

**Reporting summary.** Further information on research design is available in the Nature Research Reporting Summary linked to this article.

## Data availability

Visit data were obtained under a data confidentiality agreement from the Animal and Plant Health Agency (contact enquiries@apha.gsi.gov.uk). Honey bee import data were obtained under license from the EU TRACES database and are presented in Fig. 5 and Supplementary Table 1. The source data underlying Figs. 1a, b, 2, 3 and 6 and Supplementary Figs. 1 and 2 are all provided as a Source Data file.

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

## Acknowledgements
This work was funded jointly by BBSRC grants BB/R00482X/1 (Newcastle University) and BB/R00305X/1 (University of St Andrews) in partnership with The Bee Farmers' Association and the National Bee Unit of the Animal and Plant Health Agency.

## Author contributions
G.E.B., S.P.R. and D.J.E. conceived the study. G.E.B. and M.A.B. prepared the visit data. N.K.S., P.J.H. and P.S.M.V.W. collated adult bee samples, extracted RNA and ran RT qPCR assays. G.E.B., M.D.F.S. and S.P.R. analysed the data, and all authors wrote the manuscript.

## Competing interests
The authors declare no competing interests.
