## [Peer Review File · Nature Communications]

Reviewers' Comments:

Reviewer #1:

Remarks to the Author:

This paper aimed to determine if chronic bee paralysis virus is an emerging infectious disease of honeybees in England and Wales. By using governmental data on inspections, the authors show an exponential increase of cases in the last decade with cases that clustered spatially but not temporally, suggesting that the pathogen was spreading locally but did not persist in the same spot between years. The overall increase across the region in the last decade is convincing that this is indeed an emerging infectious disease. It is interesting that they found the highest increase in prevalence among professional beekeeper's hives in comparison to amateur beekeepers, as well as beekeepers that imported hives. The authors attempted to track the potential origin of the virus by determining what regions the hives were imported from and which were associated with the highest rates of increase. My sense is that this paper is important because the authors try to understand the large scale spatial and temporal trends of patterns in viruses in a region, along with piecing apart what practices are causing higher risks. Overall it is a well written, with clear logic and justification and results. At its heart it is a descriptive, observational paper that will be interesting to the field.

Comments

Line 37-38: Honeybee's globally have been increasing in population across that time span right?

Line 48: Add in common name for *Apis cerana* like you do for *Apis mellifera*.

Line 49: clarify that they are new diseases for the western honeybee specifically.

Line 172 – $p=0.0000$ – is a nonsense – be consistent with <0.001 – again 2 lines later – these analyses are pretty pointless. – this is against random?

Line 232-233: Could add in the aspect of a sampling bias. Are the professional beekeepers more likely to inspect and report issues in their hives since that is their main form of income, in comparison to amateurs.

Line 282: Clarification about the details and shortcomings of the disease data set. When are these health inspections occurring? On a regular time interval basis or when they are requested?

Line 299-300: how were the putative cases checked? Were there genetic tests to validate infection? What about asymptomatic cases?

Reviewer #2:

Remarks to the Author:

The manuscript "The Chronic bee paralysis: A serious emerging threat to honey bees", is an very interesting work on the epidemiology in honeybee and reads well. Using geo-statistical methods the authors present a picture of the distribution of the specific syndrome (hairless and grease) in time and space. I think this manuscript present novel epidemiological data using a geostatic approach that can be of a broad interest. I am not an expert in the geo-statistics and thus can't judge the accuracy of the methodology used and the conclusions, but from what is described the information is important and needs to be published.

However, as the data are entirely summary of records collected based on the symptoms alone without any molecular conformation of the CPPV diagnostics the conclusions need to be more careful addressed. Indeed the symptoms described fit well those of CBPV, but do they entirely exclude other possibilities? Can we really claim CBPV infection without the clear diagnostics of the CBPV viral load by RTPCR and more so it's replication? This is especially relates to the sources of the inoculum and the modes of its spread. What do we know about the spread of the virus in these countries? Are there any variants of the virus that could help us to trace the source/s? Can the differences between large and amateur beekeepers come from the fact that the former provide pollination services, thus exposing the bees to various stress factors?

As it is clear that the records of the past can't be validated by molecular methods I suggest the authors either confirm the presence of CBPV in the current samples or at least be very careful in their conclusions. I also suggest to refer to recent review about bee viruses by Grozinger & Flenniken in *Annu. Rev. Entomol.* 2019. 64:205–26.

Some small technical comments are bellow:

Fig 2 is not clear. This graph is not informative and somewhat confusing. The meaning of the distance's is not clear. In its legend line 102 remove second "cases".

Fig 5 legend is not informative enough. How exactly the risk was assessed?

L. 205 you list possible ways of transmission, what about the possibility of sexual transmission?

The authors thank the reviewers for their comments and questions and we hope our responses (in yellow) help address these.

Reviewers' comments:

Reviewer #1 (Remarks to the Author):

This paper aimed to determine if chronic bee paralysis virus is an emerging infectious disease of honeybees in England and Wales. By using governmental data on inspections, the authors show an exponential increase of cases in the last decade with cases that clustered spatially but not temporally, suggesting that the pathogen was spreading locally but did not persist in the same spot between years. The overall increase across the region in the last decade is convincing that this is indeed an emerging infectious disease. It is interesting that they found the highest increase in prevalence among professional beekeeper's hives in comparison to amateur beekeepers, as well as beekeepers that imported hives. The authors attempted to track the potential origin of the virus by determining what regions the hives were imported from and which were associated with the highest rates of increase.

My sense is that this paper is important because the authors try to understand the large scale spatial and temporal trends of patterns in viruses in a region, along with piecing apart what practices are causing higher risks. Overall it is a well written, with clear logic and justification and results. At its heart it is a descriptive, observational paper that will be interesting to the field.

Comments

Line 37-38: Honeybee's globally have been increasing in population across that time span right? It depends which data are used - Aizen and Harder (2009) report 45% increases in global honey bee populations using FAO data. The reference we chose used data derived from historic local data running to 2005 in the US and 2008 in Europe. These are however, regional rather than global, and so we have modified the text to state regional declines based on evidence presented in Ref 6.

Line 48: Add in common name for *Apis cerana* like you do for *Apis mellifera*. Added as requested

Line 49: clarify that they are new diseases for the western honeybee specifically. Both *Varroa destructor* and *Nosema ceranae* are thought to have originated from the Eastern honey bee. We have further clarified this statement.

Line 172 – $p=0.0000$ – is a nonsense – be consistent with <0.001 – again 2 lines later – these analyses are pretty pointless. – this is against random? These analyses are generalised linear models (GLMs), and so we are not sure how to address the comment '[is this] against random' because GLMs do not compare against random. We have changed the P value to $P<0.001$ and clarified the method used (Correction now Line 204).

Line 232-233: Could add in the aspect of a sampling bias. Are the professional beekeepers more likely to inspect and report issues in their hives since that is their main form of income, in comparison to amateurs. This is a valid point. Interestingly, the frequency of visits to professional beekeepers that were call-outs was about half those of amateur beekeepers, suggesting, professional beekeepers were less likely to call out an inspector. We have added text at Line 84 to highlight this feature.

Line 282: Clarification about the details and shortcomings of the disease data set. When are these health inspections occurring? On a regular time interval basis or when they are requested? This is also a valid point. In the UK, honey bee health inspections can only occur during the active beekeeping season, and so the majority of inspections are completed between April and September each year. We have provided clarity in Line 79 and produced a monthly summary of inspection in a new Supplementary Figure 2.

Line 299-300: how were the putative cases checked? Were there genetic tests to validate infection? What about asymptomatic cases? Both reviewers highlighted that we were basing our chronic bee paralysis cases on the presence of characteristic symptoms without confirming the presence of the causative organism (chronic bee paralysis virus - CBPV). We have resolved this issue by presenting CBPV data from 25 colonies that were deemed to be 'symptomatic' in the field by NBU inspectors and bee farmers, and 24 colonies that were deemed asymptomatic. We now present qualitative and quantitative data using an established CBPV RT qPCR assay on adult bees from these colonies. These additional data confirmed the presence of CBPV at high levels in symptomatic adult honey bees from the majority of disease cases, suggesting false positive rates from using survey data were low. These new data also suggest that symptoms of chronic bee paralysis are characteristic, and that symptomatic bees are a useful indicator of disease presence. We have added results, methods and discussion points for these new analyses (see Line 102 and new Figure 2 for results; Line 231 for discussion; and Line 345 for methods).

Reviewer #2 (Remarks to the Author):

The manuscript "The Chronic bee paralysis: A serious emerging threat to honey bees", is a very interesting work on the epidemiology in honeybee and reads well. Using geo-statistical methods the authors present a picture of the distribution of the specific syndrome (hairless and grease) in time and space. I think this manuscript present novel epidemiological data using a geostatic approach that can be of a broad interest. I am not an expert in the geo-statistics and thus can't judge the accuracy of the methodology used and the conclusions, but from what is described the information is important and needs to be published.

However, as the data are entirely summary of records collected based on the symptoms alone without any molecular confirmation of the CPPV diagnostics the conclusions need to be more careful addressed. Indeed the symptoms described fit well those of CBPV, but do they entirely exclude other possibilities? Can we really claim CBPV infection without the clear diagnostics of the CBPV viral load by RTPCR and more so it's replication? This is especially relates to the sources of the inoculum and the modes of its spread. What do we know about the spread of the virus in these countries? Are there any variants of the virus that could help us to trace the source/s? Can the differences between large and amateur beekeepers come from the fact that the former provide pollination services, thus exposing the bees to various stress factors? As it is clear that the records of the past can't be validated by molecular methods I suggest the authors either confirm the presence of CBPV in the current samples or at least be very careful in their conclusions. Both reviewers shared this concern. We were indeed basing our chronic bee paralysis cases on the presence of characteristic symptoms without confirming the presence of the causative organism (chronic bee paralysis virus - CBPV). As stated above, we have attempted to resolve this issue by presenting CBPV data from 25 colonies that were deemed to be 'symptomatic' in the field by NBU inspectors and bee farmers, and 24 colonies that were deemed asymptomatic. Please see above comments (see Line 102 and new Figure 2 for results; Line 231 for discussion; and Line 345 for methods).

I also suggest to refer to recent review about bee viruses by Grozinger & Flenniken in *Annu. Rev. Entomol.* 2019. 64:205–26. This review is excellent and we have added a quote at L235 to highlight the multiple interacting mechanisms that can impact viral.

Some small technical comments are below:

Fig 2 is not clear. This graph is not informative and somewhat confusing. The meaning of the distance's is not clear. In its legend line 102 remove second "cases". Figure 2 (now moved to Figure 3) is important because it highlights that chronic bee paralysis cases are spatially clustered. We have reworked the legend to better describe the figure (see Line 131).

Fig 5 legend is not informative enough. How exactly the risk was assessed? We have rewritten the legend to include the method of calculation that was previously presented in the methods section. We hope this improves clarity (see Line 214).

L. 205 you list possible ways of transmission, what about the possibility of sexual transmission? We included the possibility of vertical transmission through the queen (Ref 23), but we are not aware of any evidence to suggest CBPV is transmitted sexually.

Reviewers' Comments:

Reviewer #1:

Remarks to the Author:

The key issue with the paper has been addressed in that they have validated the diagnostics with the PCR data. I find the conclusions convincing and think this is important to the honey bee disease field.

Reviewer #2:

Remarks to the Author:

It is a revised version of an interesting manuscript on the epidemiology of honeybee disease: chronic bee paralysis.

The authors are looking at a big picture of a history and spatial distribution of a specific symptom. In my opinion it is a good approach. With one important limitation that past infection by a particular agent can not be confirmed.

In the corrected version the question raised were addressed well and in details and the current infection by CBPV was confirmed. This of course can at most imply on the past agent/s responsible for the infection.

Based on the fact that there the authors have no practical way to confirm past CBPV infection, I strongly suggest the authors to be more careful in their statements. In my opinion, the statement in the first opening sentence is too strong! I recommend to replace the word demonstrate chronic paralysis... or slightly modify the sentence and its connection with the second one, as the PCR data are relevant only for the current situation and not the past.

Minor comment: in fig 3. Y axis title needs to be more informative not just k value

Overall, the article reads well and in my opinion carries information worthy publishing

The authors thank the reviewers for their comments and questions and we hope our responses (in yellow) help address these.

Reviewers' comments:

Reviewer #1 (Remarks to the Author):

The key issue with the paper has been addressed in that they have validated the diagnostics with the PCR data. I find the conclusions convincing and think this is important to the honey bee disease field.

Thank you for the time taken to review our paper, it is much appreciated.

Reviewer #2 (Remarks to the Author):

It is a revised version of an interesting manuscript on the epidemiology of honeybee disease: chronic bee paralysis. The authors are looking at a big picture of a history and spatial distribution of a specific symptom. In my opinion it is a good approach. With one important limitation that past infection by a particular agent cannot be confirmed.

In the corrected version the question[s] raised were addressed well and in details and the current infection by CBPV was confirmed. This of course can at most imply on the past agent/s responsible for the infection.

Based on the fact that there the authors have no practical way to confirm past CBPV infection, I strongly suggest the authors to be more careful in their statements. In my opinion, the statement in the first opening sentence is too strong! I recommend to replace the word demonstrate chronic paralysis... or slightly modify the sentence and its connection with the second one, as the PCR data are relevant only for the current situation and not the past.

We have modified the wording from “demonstrate” to “indicate”, but we really do not agree with this comment. We have used visit data to screen for a very specific set of symptoms associated with chronic bee paralysis that are well detailed in the literature (for example see Ribiere et al 2010). We have then individually checked every case, to ensure the overarching description meets our structured description of the disease. We have then taken a sub-set of these cases and identified the presence of the causative organism using quantitative real-time RT PCR. We can think of no equally compelling hypothesis to explain our data than that cases we highlight represent chronic bee paralysis caused by CBPV. As such we are confident that our conclusions are robust.

Minor comment: in fig 3. Y axis title needs to be more informative not just k value.

We have added “Ripley’s k value” to the axis title.

Overall, the article reads well and in my opinion carries information worthy publishing

Thank you for the time taken to review our paper, it is much appreciated.